# Physical Activity and Perceived Physical Fitness during the COVID-19 Epidemic: A Population of 40- to 69-Year-Olds in Japan

**DOI:** 10.3390/ijerph18094832

**Published:** 2021-04-30

**Authors:** Hyuma Makizako, Shoma Akaida, Saki Shono, Ryuhei Shiiba, Yoshiaki Taniguchi, Daijo Shiratsuchi, Yuki Nakai

**Affiliations:** 1Department of Physical Therapy, School of Health Sciences, Faculty of Medicine, Kagoshima University, 8-35-1 Sakuragaoka, Kagoshima 890-8544, Japan; 2Graduate School of Health Sciences, Kagoshima University, 8-35-1 Sakuragaoka, Kagoshima 890-8544, Japan; koropitusan@gmail.com (S.A.); ssaakkii.97.618@gmail.com (S.S.); njpw527@gmail.com (R.S.); p.taniguchi0601@gmail.com (Y.T.); squall.lion8062@gmail.com (D.S.); 3Department of Rehabilitation, Tarumizu Municipal Medical Center Tarumizu Chuo Hospital, 1-140 Kinco-cho, Tarumizu, Kagoshima 891-2124, Japan; 4Department of Physical Therapy, Kagoshima Medical Professional College, 5417-1 Hirakawa, Kagoshima 891-0133, Japan; 5Department of Rehabilitation, Japan Community Health Care Organization Kumamoto General Hospital, 10-10 Tori-cho, Yatsushiro, Kumamoto 866-8660, Japan; 6Department of Mechanical Systems Engineering, Faculty of Engineering, Daiichi Institute of Technology, 1-10-2 Kokubuchuo, Kirishima, Kagoshima 899-4395, Japan; y-nakai@daiichi-koudai.ac.jp

**Keywords:** physical activity, midlife, COVID-19

## Abstract

The COVID-19 pandemic has caused an abrupt change in lifestyle for many people with restrictions, often leading to a decrease in physical activity (PA), and thus contributing to a negative perception of health status. The purpose of this study was to examine the effects of the COVID-19 epidemic on physical activity and perceived physical fitness in Japanese adults aged 40 to 69 years. Data were collected from an online survey conducted between October 19 and 28, 2020. The analytic sample consisted of 1989 Japanese adults (mean age, 50.1 ± 6.9 years; women, 38.9%) who were aged between 40 and 69 years and completed the online survey. Overall, the PA time per week decreased by 32.4% between October 2019 and April 2020. A decrease in PA time was recorded in October 2020; however, a decline of 15.5% was observed. Compared to individuals who did not perceive a decline in physical fitness, individuals who perceived declining physical fitness during the COVID-19 state of emergency demonstrated a greater decrease in PA time in April 2020 (–50.5%), and this trend continued into October 2020 (–25.0%). These findings may indicate that Japanese adults aged 40 to 69 years who perceived declining physical fitness experienced a greater decrease in physical activity.

## 1. Introduction

The COVID-19 epidemic has dramatically changed the daily lives of people worldwide. Lifestyle and health behavior including physical activity (PA) and dietary habits during the COVID-19 lockdown have been investigated [1,2]. The impact of the COVID-19 pandemic on dietary intake, PA, and sedentary behavior among different groups such as university students, elite para-athletes, community-dwelling older people, and patients with type 2 diabetes have been examined [1,3,4,5]. For instance, the COVID-19 state of emergency or lockdown has led to PA restrictions worldwide [6]. In Japan, a state of emergency was declared in seven prefectures, including Tokyo, on April 7, 2020 and extended to the entire country on April 16 [7]. On May 25, the state of emergency was lifted nationwide for the first time in about a month and a half. Although the declaration of a state of emergency is not legally binding, the prefectural governors of the target areas can request that residents refrain from going out as part of a collective effort to reduce infection, except when doing so is necessary for the maintenance of daily life. Among Japanese older adults aged ≥ 65 years, the PA time per week decreased by 65 min (–26.5%) from January to April 2020 [8]. If PA rapidly decreases for an extended period, it may have a negative impact on health status.

Lower midlife PA is associated with a higher risk of cardiovascular diseases [9], metabolic syndrome [10], and neurodegenerative diseases [11]. PA in midlife is also related to perceived health [12]. Perceived health status affects negative health-related outcomes, such as mortality, functional impairment, and dementia [13,14,15]. Lifestyle changes during the state of emergency may affect perceived health status, which negatively affects future health conditions. A previous longitudinal study investigated changes in PA prior to lockdown restrictions being imposed, and across three time periods: pre-, during, and post-lockdown [16]. The results showed that vigorous and moderate intensity PA were significantly lower during and post-lockdown compared to pre-lockdown in those individuals who had been highly active pre-lockdown [16]. More than 40% of community-dwelling Japanese old-old adults perceived a decline their physical fitness during the COVID-19 state of emergency [17]. Although several studies have reported decreased PA due to the COVID-19 epidemic [18], few have determined its impact on perceived health and PA recovery. Clear associations between decreased PA due to the COVID-19 epidemic and poor perceived physical health status may exist.

The purpose of this study was to examine the effect of the first wave of the COVID-19 state of emergency on physical activity and perceived physical fitness in Japanese adults aged 40 to 69 years. We examined the PA time across three time periods: pre- (October 2019), during (April 2020) and post- (October 2020) COVID-19 state of emergency and compared those between Japanese adults aged 40 to 69 years who perceived declining physical fitness during the COVID-19 state of emergency and those who did not.

## 2. Materials and Methods

### 2.1. Study Sample

Data for this study were collected from an online survey panel administered through the sampling of Y cloud systems among Japanese adults. The Y cloud system is a crowdsourcing service launched by Yahoo Japan Corporation, Inc. (Tokyo, Japan) in 2013. From 19 to 28 October 2020, 3048 Japanese adults completed the online survey. The inclusion criterion was adults aged 40 to 69 years. Responders who reported a history of stroke, Parkinson’s disease, dementia, depression, and/or neurological disorders, or who gave the wrong answer to a question (choosing a specific option from multiple choices) to identify fraudulent responses were excluded. Responders who reported more than 960 min/day or 0 min/day of total PA time and more than a tenfold change (increasing or decreasing) of PA were also excluded. Finally, data from 1986 Japanese adults aged 40 to 69 years were analyzed. This study was conducted in accordance with the guidelines proposed by the Declaration of Helsinki, and the study protocol was reviewed and approved by the Ethics Committee of the Faculty of Medicine, Kagoshima University (#200101).

### 2.2. Assessment of PA

The abbreviated version of the International Physical Activity Questionnaire (IPAQ) [19] which consists of three-dimensional activity items—activity intensity level (light, moderate, and vigorous intensity), activity frequency per week, and activity time per day—was used to assess PA. Participants were asked to report their PA during three specific time periods: (1) October 2019, before the COVID-19 epidemic; (2) April 2020, during the first wave of the COVID-19 state of emergency; and (3) October 2020, after the COVID-19 state of emergency. Thus, they were asked to recall October 2019 and April 2020, as well as to report their current situation in October 2020. Following the guidelines for data processing and analysis of the IPAQ, only values of 10 or more minutes of PA were included in the calculation of summary scores. Responses of less than 10 min (and their associated days) were re-coded to ‘zero’. Additionally, activity time variables of each level exceeding ‘3 h’ or ‘180 min’ were truncated to be equal to ‘180 min’ in a new variable. We determined the PA time (minutes/week) as added values for each activity level, which were then multiplied by activity frequency per week and activity time per day (minutes) at each activity level [8].

### 2.3. Assessment of Perceived Declining Physical Fitness

We investigated the participants’ perceived decline in physical fitness during the COVID-19 state of emergency. Participants were asked to answer “yes” or “no” to the following question: “Do you perceive a decline in physical fitness after the state of emergency?” Participants who answered “yes” to this question were classified as perceiving a decline in their physical fitness. Incidentally, in this survey, respondents were asked about their physical fitness self-perception during the COVID-19 epidemic prior to completing the IPAQ.

### 2.4. Demographic Variables

Several variables such as age, gender, education, living alone, and living area were collected in the current study. The living area was classified as a special precaution area due to the COVID-19 pandemic. In Japan, 13 prefectures, including Tokyo, Kanagawa, Saitama, Chiba, Osaka, Hyogo, Fukuoka, Hokkaido, Ibaraki, Ishikawa, Gifu, Aichi, and Kyoto, were identified as areas of special precaution based on the situation in April 2020.

### 2.5. Statistical Analysis

PA time is presented as a median, with an interquartile range (IQR). The IPAQ guidelines state that physical activity is non-normally distributed in many populations and suggest reporting medians [20]. Therefore, we conducted the analysis using non-parametric tests, as appropriate. Changes in PA time were tested using Friedman’s test for three time points, in October 2019, April 2020, and October 2020, overall and for each group, divided by perceived declining or non-declining physical fitness. Changes in PA time were also compared with October 2019 (before the COVID-19 epidemic) and tested using the Wilcoxon rank-sum test overall and for each group, divided by perceived declining or non-declining physical fitness. The changes in PA time also compared with October 2019 (before the COVID-19 epidemic) were tested using the Wilcoxon rank-sum test in overall and each group divided by perceived declining or non-declining physical fitness. Comparisons of demographic variables and PA time between perceived and non-perceived decline in physical fitness were tested using the paired *t*-test (for age), Mann–Whitney U test (for PA time), or chi-square test (for proportion). All analyses were conducted using IBM SPSS Statistics 26.0 (IBM Japan Tokyo, Japan). The level of statistical significance was set at *p* < 0.05.

## 3. Results

The demographic variables and PA time in October 2019, April 2020, and October 2020 for each age group (40–49, 50–59, and 60–69 years) are presented in Table 1. The mean (± standard deviation) age of the participants was 50.1 ± 6.9 years and 38.9% were women. Overall, the PA time decreased (*p* < 0.001) in April 2020 (median [IQR], 240 [80–540]) and October 2020 (300 [120–600]) compared to October 2019 (355 [150–660]).

Table 2 illustrates the comparisons between participants who perceived a decline in physical fitness and those who did not in terms of demographic variables and PA time. Of the overall participants, 671 (33.8%) perceived a decline in physical fitness during the COVID-19 state of emergency. There were no significant differences between those who perceived a decline in physical fitness and those who did not in terms of education and who they lived with, but there was a higher rate of women and those who lived in special precaution areas among participants who perceived a decline in their physical fitness. Although there was no significant difference in PA time in October 2019 between those who had perceived a decline in physical fitness and those who had not, there was a significant difference in PA time between the former (180 [60–480]) and the latter (270 [100–600]) in April 2020 (*p* < 0.01). In October 2020, this difference was not evident, but there was a slight decrease in the group with a perceived decline in physical fitness (270 [90–580]) compared with the group with no perceived decline in physical fitness (300 [120–620]) (*p* = 0.48).

Figure 1 shows the changes among the three time points, in October 2019, April 2020, and October 2020. Overall, there were significant differences in PA time among the three time points (*p* < 0.001). A significant 32.4% decline in PA time in April 2020 (*p* < 0.01) was observed, compared with October 2019. The decreased PA time improved in October 2020; however, the median PA time still decreased (–15.5%), compared with October 2019 (*p* < 0.01). PA time change results for the perceived non-declining and declining physical fitness groups are also illustrated in Figure 1. Participants who perceived declining physical fitness during the COVID-19 state of emergency showed more than a 50% decrease in PA time in April 2020 (–50.5%) (*p* < 0.01), and that decline remained (to a lesser degree) in October 2020 (–25.0%) (*p* < 0.01), compared with October 2019. Conversely, in participants who did not perceive declining physical fitness during the COVID-19 state of emergency, PA time during the COVID-19 state of emergency (April 2020) (–20.6%) and afterward (October 2020) (–11.8%) showed a smaller decline; however, those decreased PA times were still at a significant level (*p* < 0.01).

## 4. Discussion

This study examined PA time across three time periods: pre- (October 2019), during (April 2020) and post- (October 2020) COVID-19 state of emergency and compared PA times in those three time periods between Japanese adults aged 40 to 69 years who perceived a decline in physical fitness and those who did not. The current study confirmed that weekly PA time decreased from October 2019 to April 2020 by approximately 30%. This decrease in PA time improved in October 2020; however, a decline of approximately 15% was still observed. Specifically, the decrease in PA time in participants who perceived declining physical fitness during the COVID-19 state of emergency was remarkable, with declines of approximately 50% and 25% in April and October 2020, respectively. Those results would indicate secondary effects of the COVID-19 epidemic on our health, and continuously decreasing PA may contribute to negative health outcomes.

A previous study indicated that step counts decreased in the period after COVID-19, and differences were observed between regions, likely reflecting regional variation in COVID-19 timing, regional enforcement, and behavior change [6]. Among Japanese community-dwelling older adults, although the total amount of time devoted to PA in April 2020 (during the first wave of the COVID-19 pandemic) had significantly decreased from that in January 2020 (before the COVID-19 pandemic), PA time in June 2020 had recovered to the same level as before the COVID-19 pandemic (January 2020) [21]. The current study revealed that approximately 30% of the PA time decreased during the first wave of the COVID-19 pandemic among Japanese adults aged 40 to 69 years. The decrease in PA time in October 2020 was still approximately 15% lower than that before the COVID-19 pandemic (October 2019). Further long-term follow-up observation and examination of the effects of PA time changes on various health outcomes are needed.

Several previous studies examined that PA in midlife using the IPAQ to determine the prevalence of physical inactivity [22], and to examine the association of PA with neighborhood walkability [23] and the risk of health problems such as stroke [24], depression [25], and cognitive impairment [26]. Nearly all studies indicated that a lower PA in midlife had a negative impact on current and future health. During the COVID-19 lockdown, an online survey of a young cohort assessing its effect on PA levels showed an increase in sedentary behavior [27]. On the other hand, middle-aged adults with high and moderate PA levels had significantly higher life satisfaction and happiness than those with low PA levels [28]. Thus, to avoid negative impacts on health, unfavorable changes in PA time due to the COVID-19 state of emergency must reverse and recover to the level before the COVID-19 pandemic.

In this study, 33.8% (mean age 50.1 years) of participants perceived a decline in their physical fitness during the COVID-19 state of emergency. In general, self-rated health tends to deteriorate in older adults compared to middle-aged adults [29]. Poor self-rated health and physical fitness among middle-aged adults are associated with future negative health outcomes [30,31,32,33] and PA levels affect self-rated health status [34,35,36]. The current study identified an association between PA time and poor self-rated physical fitness during the COVID-19 state of emergency among Japanese adults aged 40 to 69 years, and those with perceived poor self-rated physical fitness showed less recovery of their PA after the first wave of the COVID-19 epidemic. In addition, a short-term reduction in steps resulted in a significant loss of leg fat-free mass [37]. Short-term physical inactivity also requires attention, and early improvement in PA is important [37]. Thus, the maintenance or minimization of changes in PA time due to the COVID-19 outbreak is very important, and if the PA decreases, it should be recovered as soon as possible.

One strength of this study is that it collected a representative sample of Japanese adults aged 40 to 69 years in the national area. Given the importance of PA and how affected it was by the restrictions imposed by the COVID-19 state of emergency, the results of this study will be important for planning and developing appropriate strategies and policies in the face of a pandemic. In addition to its strengths, some limitations should be considered when interpreting the results of the current study. Frist, participants were asked to recall their PA one year before the survey, in October 2019. Thus, it is possible that their subjective PA times could have been either underestimated or overestimated. Second, changes in PA time were assessed using the IPAQ and analyzed using non-parametric tests, following the IPAQ guidelines. We could not control for confounders, such as age, gender, and income. The decline in physical fitness is self-rated and not an objective measurement of physical fitness. Subjective PA levels may be over- or underestimated [38]. Furthermore, confounding factors between self-rated physical weakness and PA were not considered. In addition, the two groups were determined according to their perception of a decline in physical fitness at the end of the survey period, October 2020; as they were not divided according to objective measures, several April 2020 results might be different. For instance, some members of the “non-declining” group may have identified as members of the “declining” group if they had been asked in April 2020. For instance, some individuals of the “non-declining” group, if they were asked in April 2020, would answer as a “declining” group member. Therefore, some individuals may believe that there is a self-perception threshold regarding one’s physical fitness status, despite any given individual’s knowledge that they were, indeed, performing less PA. In future research, these points could contribute to understanding the associations between PA levels and self-perceived fitness. Third, participants were recruited from specific Internet service registrants.

## 5. Conclusions

The PA time per week for Japanese adults aged 40 to 69 years decreased by 32.4% between October 2019 and April 2020 with a continued 15.5% median decline in October 2020. The decrease in PA time in participants who perceived a decline in their physical fitness during the COVID-19 state of emergency was remarkable in April 2020 and October 2020. These findings indicate that Japanese adults who perceived a decline in their physical fitness experienced a greater decrease in PA and found it difficult to return to the same level of PA as the past year.

## Figures and Tables

**Figure 1 ijerph-18-04832-f001:**
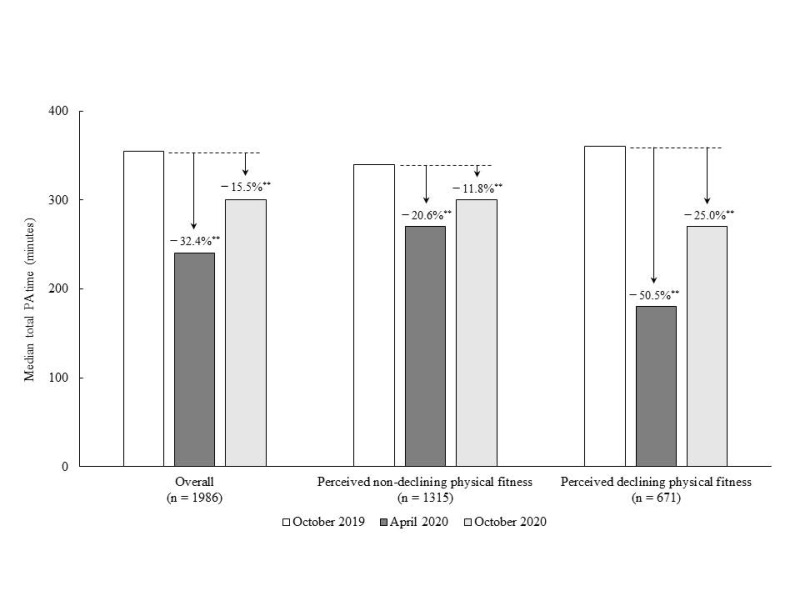
Changes in PA time in April and October 2020 compared with October 2019 in participants with and without a perceived decline in physical fitness. **: statistical significance (*p* < 0.01) compared with October 2019.

**Table 1 ijerph-18-04832-t001:** Demographic characteristics and physical activity (PA) time of the participants in each age groups.

	Overall (*n* = 1986)	40–49 Years (*n* = 1046)	50–59 Years (*n* = 712)	60–69 Years (*n* = 228)
Age, years	50.1 ± 6.9	44.8 ± 2.8	55.7 ± 2.8	63.4 ± 2.9
Women, *n* (%)	773 (38.9%)	453 (43.3%)	256 (36.0%)	64 (28.1%)
Education, *n* (%)				
Master/doctorate degree	110 (5.5%)	55 (5.3%)	38 (5.3%)	17 (7.5%)
Bachelor’s degree	1011 (50.9%)	535 (51.1%)	345 (48.5%)	131 (57.5%)
Professional degree	311 (15.7%)	180 (17.2%)	105 (14.7%)	26 (11.4%)
High school graduate	465 (23.4%)	232 (22.2%)	185 (26.0%)	48 (21.1%)
Others	89 (4.5%)	44 (4.2%)	29 (4.1%)	6 (2.6%)
Living alone, *n* (%)	368 (18.5%)	201 (19.2%)	129 (18.1%)	38 (16.7%)
Living area, *n* (%)				
Special precaution areas due to COVID-19 pandemic	1415 (71.2%)	731 (69.9%)	532 (74.7%)	152 (66.7%)
Perceived declining physical fitness (yes), *n* (%)	671 (33.8%)	364 (34.8%)	227 (31.9%)	80 (35.1%)
PA time, minutes				
October-2019, median (IQR)	355 (150–660)	343 (150–700)	335 (150–630)	360 (161–685)
April-2020, median (IQR)	240 (80–540)	240 (60–560)	233 (80–490)	300 (120–600)
October-2020, median (IQR)	300 (120–600)	300 (100–600)	298 (120–600)	300 (120–630)

Abbreviations: PA, physical activity; IQR., interquartile range.

**Table 2 ijerph-18-04832-t002:** Comparison between participants with and without a perceived decline in physical fitness.

	No Perceived Decline in Physical Fitness (*n* = 1315)	Perceived Decline in Physical Fitness (*n* = 671)	*p*
Age, years	50.2 ± 6.9	50.0 ± 6.9	0.57
Women, *n* (%)	469 (35.7%)	304 (45.3%)	<0.01
Education (Bachelor/master/doctorate degree), *n* (%)	587 (44.6%)	278 (41.4%)	0.17
Living alone, *n* (%)	237 (18.0%)	131 (19.5%)	0.42
Living area, *n* (%)			
Special precaution areas due to COVID-19 pandemic	913 (69.4%)	502 (74.8%)	0.01
PA time, minutes			
October 2019, median (IQR)	340 (140–660)	360 (150–700)	0.43
April 2020, median (IQR)	270 (100–600)	180 (60–480)	<0.01
October 2020, median (IQR)	300 (120–620)	270 (90–580)	0.48

Abbreviations: PA, physical activity; IQR., interquartile range.

## Data Availability

Data sharing not applicable.

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
