# Peer review of "Physical Activity and Perceived Physical Fitness during the COVID-19 Epidemic: A Population of 40- to 69-Year-Olds in Japan"

_ijerph, 2021, doi:10.3390/ijerph18094832_

Round 1

Reviewer 1 Report

General comments.

The aim of this study was to examine the associations between changes in physical activity and perceived declining physical fitness in middle-aged adults. The findings showed the effects of the COVID-19 epidemic on physical activity and health in Japan.

This paper has the important role for findings of effects of COVID-19 epidemic on lifestyle (i.e. physical activity) and health. However, I have some concerns before publication.

  1. The main focus should be the association between physical activity and perceived physical fitness because of the title and purpose; however, this paper has few discussions about this topic. This paper mainly focused the decline of physical activity in the discussion. You should change the title and purpose OR please discuss the association of two variables.
  2. If you found the association, could you use the other statistical analysis controlling for confounders (age, gender, income etc.). Furthermore, please use the statistical analysis for changes in physical activity in group. Although you showed only median change (Figure 1), you used the words to guide your thoughts, mislead, “decline, not return, unfavorable change, improved”. Why did you use the median change not the other analysis like two way ANOVA and so on?
  3. All contents were asked in same period. If there were the question about perceived physical fitness before the physical activity, did participants who answered the more decline physical activity get easier to answer to “yes” to the question about perceived physical fitness. Therefore, is there possibility that this research misled participants who perceived the decreases in physical activity answered to the perceived physical fitness. Please describe the validity of this research.

Specific comments.

Title. I felt that the primary focus of this paper was the effect of the COVID-19 epidemic on the physical activity and perceived physical fitness, not the association between two variables because there is few discussion about the association. Therefore, I suggest you to change the title.

l26. You used the “hyphen” as “minus sign”. Please correct it.

ll.30-31. You overstated the findings “could not return to …” because the findings showed the mean difference and no significant difference between perceived and non-perceived declining physical fitness groups. I suggest you change the sentence or doing other statistical methods.

l76. Were the adults aged from 65 to 69 years categorized as the middle-aged not older?

ll70-121. Please describe the precaution for COVID-19 epidemic in Japan (duration of self-quarantine, how much the legal force, and etc.)

l135. Please delete “were” in this sentence.

l149-160. Although current study did not analysis the change in physical activity from pre COVID-19 and recover from April to October 2020, these paragraphs indicated no recovery in adults with perceived physical fitness. Moreover, Table 2 showed the no significant difference of physical activity between groups in October 2020. Please describe more detail about why you thought.

l173. Please correct the value, change from 50.0% to 50.5%.

ll206-207. Please add the reference.

Author Response

Manuscript ID: ijerph-1187899
Title: Associations of physical activity changes and perceived declining physical fitness during the COVID-19 state of emergency among Japanese middle-aged adults

Reviewer 1

The aim of this study was to examine the associations between changes in physical activity and perceived declining physical fitness in middle-aged adults. The findings showed the effects of the COVID-19 epidemic on physical activity and health in Japan.

This paper has the important role for findings of effects of COVID-19 epidemic on lifestyle (i.e. physical activity) and health. However, I have some concerns before publication.

Response

We appreciate your helpful and insightful comments. The revised manuscript was reviewed by a native English speaker to fix syntactical and spelling errors. Please see our responses to your specific comments below.

The main focus should be the association between physical activity and perceived physical fitness because of the title and purpose; however, this paper has few discussions about this topic. This paper mainly focused the decline of physical activity in the discussion. You should change the title and purpose OR please discuss the association of two variables.

Response

I recognize that your points are particularly important and appreciate your suggestions. After careful consideration, we decided to change the title and purpose.

Location in the text

Title

Effects of the COVID-19 epidemic on physical activity and perceived physical fitness: a population of 40–69-year-olds in Japan

Abstract

The purpose of this study was to examine the effects of the COVID-19 epidemic on physical activity and perceived physical fitness in Japanese adults aged 40–69 years.

Introduction

The purpose of this study was to examine the effect of the first wave of the COVID-19 state of emergency on physical activity and perceived physical fitness in Japanese adults aged 40–69 years.

If you found the association, could you use the other statistical analysis controlling for confounders (age, gender, income etc.). Furthermore, please use the statistical analysis for changes in physical activity in group. Although you showed only median change (Figure 1), you used the words to guide your thoughts, mislead, “decline, not return, unfavorable change, improved”. Why did you use the median change not the other analysis like two-way ANOVA and so on?

Response

Thank you for your insightful comments. I agree with your suggestions regarding other analyses, like two-way ANOVA and another statistical analysis controlling for confounders. In this study, we followed the “Guidelines for Data Processing and Analysis of IPAQ (2005)” to analyze physical activity. According to the guidelines, given the non-normal distribution of energy expenditure in many populations, it is suggested that the continuous indicator be presented as median minutes/week or median MET-minutes/week, rather than as a mean. Therefore, we conducted the analysis using non-parametric tests to assess the medians. Several sentences were added in the Methods and Limitations sections regarding those points.

Location in the text

Methods (Statistical analysis)

PA time is presented as a median with an interquartile range (IQR). The IPAQ guidelines state that physical activity is non-normally distributed in many populations and suggest reporting medians [7]. Therefore, we conducted the analysis using non-parametric tests, as appropriate. Changes in PA time were tested using Friedman’s test for three time points, in October 2019, April 2020, and October 2020, overall and for each group, divided by perceived declining or non-declining physical fitness. Changes in PA time were also compared with October 2019 (before the COVID-19 epidemic) and tested using the Wilcoxon rank-sum test overall and for each group, divided by perceived declining or non-declining physical fitness.

Discussion (limitation)

Second, changes in PA time were assessed using the IPAQ and analyzed using non-parametric tests, following the IPAQ guidelines. We could not control for confounders, such as age, gender, and income.

All contents were asked in same period. If there were the question about perceived physical fitness before the physical activity, did participants who answered the more decline physical activity get easier to answer to “yes” to the question about perceived physical fitness. Therefore, is there possibility that this research misled participants who perceived the decreases in physical activity answered to the perceived physical fitness. Please describe the validity of this research.

Response

I appreciate your insights. We recognize the importance of the points you have suggested. In fact, all responses were collected in the same period. We added details pertaining to question sequence, and several related sentences were added in the Limitations section.

Location in the text

Methods

Incidentally, in this survey, respondents were asked about their physical fitness self-perception during the COVID-19 epidemic prior to completing the IPAQ.

Discussion (limitation)

In addition, the two groups were determined according to their perception of a decline in physical fitness at the end of the survey period, October 2020; because they were not divided according to objective measures, several April 2020 results might be different. For instance, some members of the “non-declining” group may have identified as members of the “declining” group, if they had been asked in April 2020.

Specific comments.

Title. I felt that the primary focus of this paper was the effect of the COVID-19 epidemic on the physical activity and perceived physical fitness, not the association between two variables because there is few discussion about the association. Therefore, I suggest you to change the title.

Response

After careful consideration, we changed the title accordingly.

Location in the text

Title

Effects of the COVID-19 epidemic on physical activity and perceived physical fitness: a population of 40–69-year-olds in Japan

l26. You used the “hyphen” as “minus sign”. Please correct it.

Response

We corrected this throughout the manuscript. Thank you.

Location in the text

Entire manuscript

ll.30-31. You overstated the findings “could not return to …” because the findings showed the mean difference and no significant difference between perceived and non-perceived declining physical fitness groups. I suggest you change the sentence or doing other statistical methods.

Response

I appreciate your suggestions. Additional statistical analyses were performed to test for within-group differences between time points. As you mentioned, there was no significant difference between the perceived declining and non-declining physical fitness groups in October 2020. Both groups showed significant PA time decreases. We therefore changed the relevant sentences in the Abstract. Several sentences were added in the revised manuscript regarding these points.

Location in the text

Abstract

Compared to individuals who did not perceive a decline in physical fitness, individuals who perceived declining physical fitness during the COVID-19 state of emergency demonstrated a greater decrease in PA time in April 2020 (–50.5%), and this trend continued into October 2020 (–25.0%). These findings may indicate that Japanese adults aged 40–69 years who perceived declining physical fitness experienced a greater decrease in physical activity.

l76. Were the adults aged from 65 to 69 years categorized as the middle-aged not older?

Response

As you mentioned, people aged 65–69 years should be categorized as elderly, not middle-aged. The title and several sentences were revised accordingly.

Location in the text

Title

Effects of the COVID-19 epidemic on physical activity and perceived physical fitness: a population of 40–69-year-olds in Japan

Methods

The inclusion criteria included middle-aged adults who were 40 to 69 years old.

Entire manuscript

ll70-121. Please describe the precaution for COVID-19 epidemic in Japan (duration of self-quarantine, how much the legal force, and etc.)

Response

I appreciate your comments. The state of emergency in Japan was implemented for about a month and a half, from April 7 to May 25, 2020. During the declaration of a state of emergency, the prefectural governors of the targeted areas can request that residents refrain from going out, except when it is necessary for the maintenance of daily life. Governors can also make requests or give instructions regarding school closure and the restricted use of public facilities. The introductory text has been revised accordingly.

Location in the text

Introduction

In Japan, a state of emergency was declared in seven prefectures, including Tokyo, on April 7, 2020 and extended to the entire country on April 16 [7]. On May 25, the state of emergency was lifted nationwide for the first time in about a month and a half. Although the declaration of a state of emergency is not legally binding, the prefectural governors of the target areas can request that residents refrain from going out as part of a collective effort to reduce infection, except when doing so is necessary for the maintenance of daily life.

l135. Please delete “were” in this sentence.

Response

This sentence was revised. Thank you.

Location in the text

Results

Of the overall participants, 671 (33.8%) participants were perceived declining physical fitness during the COVID-19 state of emergency.

l149-160. Although current study did not analysis the change in physical activity from pre COVID-19 and recover from April to October 2020, these paragraphs indicated no recovery in adults with perceived physical fitness. Moreover, Table 2 showed the no significant difference of physical activity between groups in October 2020. Please describe more detail about why you thought.

Response

I appreciate your insights and agree that those statements were overstated. Additional statistical analyses were performed to test within-group differences between time points. Several sentences were revised regarding these points.

Location in the text

Methods (Statistical analysis)

PA time is presented as a median with an interquartile range (IQR). The IPAQ guidelines state that physical activity is non-normally distributed in many populations and suggest reporting medians [20]. Therefore, we conducted the analysis using non-parametric tests, as appropriate. Changes in PA time were tested using Friedman’s test for three time points, in October 2019, April 2020, and October 2020, overall and for each group, divided by perceived declining or non-declining physical fitness. Changes in PA time were also compared with October 2019 (before the COVID-19 epidemic) and tested using the Wilcoxon rank-sum test overall and for each group, divided by perceived declining or non-declining physical fitness.

Results

Figure 1 shows the changes among the three time points, in October 2019, April 2020, and October 2020. Overall, there were significant differences in PA time among the three time points (p < 0.001). A significant 32.4% decline in PA time in April 2020 (p < 0.01) was observed, compared with October 2019. The decreased PA time improved in October 2020; however, the median PA time still decreased (–15.5%), compared with October 2019 (p < 0.01). PA time change results for the perceived non-declining and declining physical fitness groups are also illustrated in Figure 1. Participants who perceived declining physical fitness during the COVID-19 state of emergency showed more than a 50% decrease in PA time in April 2020 (–50.5%) (p < 0.01), and that decline remained (to a lesser degree) in October 2020 (–25.0%) (p < 0.01), compared with October 2019. Conversely, in participants who did not perceive declining physical fitness during the COVID-19 state of emergency, PA time during the COVID-19 state of emergency (April 2020) (–20.6%) and afterward (October 2020) (–11.8%) showed a smaller decline; however, those decreased PA times were still at a significant level (p < 0.01).

l173. Please correct the value, change from 50.0% to 50.5%.

Response

We revised the sentence according to your suggestion. Thank you.

Location in the text

Discussion

Specifically, the decrease in PA time in participants who perceived declining physical fitness during the COVID-19 state of emergency was remarkable, with declines of approximately 50% and 25% in April and October 2020, respectively.

ll206-207. Please add the reference.

Response

We added the reference there. Thank you.

Location in the text

Discussion

In addition, a short-term reduction in steps has shown significant loss of leg-fat free mass [37].

Reviewer 2 Report

This is an interesting study aiming to assess the association between changes in physical activity and perceived physical fitness during the COVID 19 pandemic. My major concern is the use of retrospective subjective measurements of physical activity. The manuscript also needs language editing.

Major comments

The introduction needs to describe the restrictions that were imposed on the people of Japan during the pandemic and when the restrictions started were adjusted and lifted etc. This is important to describe as it has relevance for the opportunities to be physical active.

The result section has nice tables and the figures are easy to understand and summarise the results in an appropriate manner. However, the text in the result section is generally difficult to read and to understand. Please revise and aim to improve the language.

The discussion section is difficult to follow, and I don’t always understand what you are writing. I would also benefit from a better structure.

After the first paragraph where the main findings are summarised, I suggest you discuss your results before comparing them to the literature.

The strength and limitation section address both the generalisability of the sample and the weaknesses related to subjective measurements of physical activity, but his needs to be strengthened substantially.

Minor comments

I suggest not using decimals for present thorough the manuscript as the decimals decrease the readability and do not provide any extra information when the differences are large in the first place.

If you describe in the statistical section how you present your data e.g. “total PA time is presented as median with inter quartile range” you can omit the description from the result and discussion text and it will be much easier to read.  

Figure 1: Can you illustrate statistically significant differences between time points within the  groups and difference between the sub groups at the same time points?

Be consistent in terminology. Use “PA time” or total “PA time”

In the discussion you may write “a decline of x%” instead of “– x% median change”

Author Response

Manuscript ID: ijerph-1187899
Title: Associations of physical activity changes and perceived declining physical fitness during the COVID-19 state of emergency among Japanese middle-aged adults

Reviewer 2

This is an interesting study aiming to assess the association between changes in physical activity and perceived physical fitness during the COVID 19 pandemic. My major concern is the use of retrospective subjective measurements of physical activity. The manuscript also needs language editing.

Response

We appreciate your helpful and insightful comments. The revised manuscript was reviewed by a native English speaker to fix syntactical and spelling errors. Please see our responses to your specific comments below.

Major comments

The introduction needs to describe the restrictions that were imposed on the people of Japan during the pandemic and when the restrictions started were adjusted and lifted etc. This is important to describe as it has relevance for the opportunities to be physical active.

Response

The state of emergency in Japan was implemented for about a month and a half, from April 7 to May 25, 2020. During the declaration of a state of emergency, the prefectural governors of the targeted areas can request that residents refrain from going out, except when it is necessary for the maintenance of daily life. Governors can also make requests or give instructions regarding school closure and the restricted use of public facilities. The introductory text has been revised accordingly.

Location in the text

Introduction

In Japan, a state of emergency was declared in seven prefectures, including Tokyo, on April 7, 2020 and extended to the entire country on April 16 [7]. On May 25, the state of emergency was lifted nationwide for the first time in about a month and a half. Although the declaration of a state of emergency is not legally binding, the prefectural governors of the target areas can request that residents refrain from going out as part of a collective effort to reduce infection, except when doing so is necessary for the maintenance of daily life.

The result section has nice tables and the figures are easy to understand and summarise the results in an appropriate manner. However, the text in the result section is generally difficult to read and to understand. Please revise and aim to improve the language.

Response

I appreciate your comments. To facilitate greater ease of comprehension, we revised several sentences in the Results section, especially those discussing PA time.

Location in the text

Results

Figure 1 shows the changes among the three time points, in October 2019, April 2020, and October 2020. Overall, there were significant differences in PA time among the three time points (p < 0.001). A significant 32.4% decline in PA time in April 2020 (p < 0.01) was observed, compared with October 2019. The decreased PA time improved in October 2020; however, the median PA time still decreased (–15.5%), compared with October 2019 (p < 0.01). PA time change results for the perceived non-declining and declining physical fitness groups are also illustrated in Figure 1. Participants who perceived declining physical fitness during the COVID-19 state of emergency showed more than a 50% decrease in PA time in April 2020 (–50.5%) (p < 0.01), and that decline remained (to a lesser degree) in October 2020 (–25.0%) (p < 0.01), compared with October 2019. Conversely, in participants who did not perceive declining physical fitness during the COVID-19 state of emergency, PA time during the COVID-19 state of emergency (April 2020) (–20.6%) and afterward (October 2020) (–11.8%) showed a smaller decline; however, those decreased PA times were still at a significant level (p < 0.01).

The discussion section is difficult to follow, and I don’t always understand what you are writing. I would also benefit from a better structure.

Response

We revised the Discussion section for clarity and greater ease of comprehension.  Additionally, the revised manuscript was reviewed by a native English speaker.

Location in the text

Entire manuscript

After the first paragraph where the main findings are summarised, I suggest you discuss your results before comparing them to the literature.

Response

I appreciate your suggestions. We discussed the present study’s results before comparing our findings to the literature.

Location in the text

Discussion

Those results would indicate secondary effects of the COVID-19 epidemic on our health, and continuously decreasing PA may contribute to negative health outcomes.

The strength and limitation section address both the generalisability of the sample and the weaknesses related to subjective measurements of physical activity, but his needs to be strengthened substantially.

Response

We reconsidered several sentences in the strength and limitation section according to your suggestions.

Location in the text

Discussion (strength and limitation)

Second, changes in PA time were assessed using the IPAQ and analyzed using non-parametric tests, following the IPAQ guidelines. We could not control for confounders, such as age, gender, and income.

Discussion (strength and limitation)

In addition, the two groups were determined according to their perception of a decline in physical fitness at the end of the survey period, October 2020; because they were not divided according to objective measures, several April 2020 results might be different. For instance, some members of the “non-declining” group may have identified as members of the “declining” group, if they had been asked in April 2020.

Minor comments

I suggest not using decimals for present thorough the manuscript as the decimals decrease the readability and do not provide any extra information when the differences are large in the first place.

Response

We reworded the title according to your suggestion, especially in the first paragraph of the Discussion section.

Location in the text

Discussion

The current study confirmed that weekly PA time decreased from October 2019 to April 2020 by approximately 30%. This decrease in PA time improved in October 2020; however, a decline of approximately 15% was still observed. Specifically, the decrease in PA time in participants who perceived declining physical fitness during the COVID-19 state of emergency was remarkable, with declines of approximately 50% and 25% in April and October 2020, respectively.

If you describe in the statistical section how you present your data e.g. “total PA time is presented as median with inter quartile range” you can omit the description from the result and discussion text and it will be much easier to read. 

Response

I appreciate your suggestions. We revised the sentences accordingly.

Location in the text

Methods (Statistical analysis)

PA time is presented as a median, with an interquartile range (IQR).

Results

Overall, there was a significant decrease in PA time in April 2020 (median [interquartile range (IQR)], 240 [80 to 540]) when compared to October 2019 (median [IQR], 355 [150 to 660]) (P < 0.001). There was also a decrease in PA time in October 2020 (median [IQR], 300 [120 to 600]) (P < 0.001).

Figure 1: Can you illustrate statistically significant differences between time points within the groups and difference between the subgroups at the same time points?

Response

I appreciate your suggestions. Additional statistical analyses were performed and added to the results in the text and Figure 1.

Location in the text

Methods (Statistical analysis)

Changes in PA time were tested using Friedman’s test for three time points, in October 2019, April 2020, and October 2020, overall and for each group, divided by perceived declining or non-declining physical fitness. Changes in PA time were also compared with October 2019 (before the COVID-19 epidemic) and tested using the Wilcoxon rank-sum test overall and for each group, divided by perceived declining or non-declining physical fitness.

Results

Figure 1 shows the changes among the three time points, in October 2019, April 2020, and October 2020. Overall, there were significant differences in PA time among the three time points (p < 0.001). A significant 32.4% decline in PA time in April 2020 (p < 0.01) was observed, compared with October 2019. The decreased PA time improved in October 2020; however, the median PA time still decreased (–15.5%), compared with October 2019 (p < 0.01). The PA time change results for the perceived non-declining and declining physical fitness groups are also illustrated in Figure 1. Participants who perceived declining physical fitness during the COVID-19 state of emergency showed more than a 50% decrease in PA time in April 2020 (–50.5%) (p < 0.01), and that decline remained (to a lesser degree) in October 2020 (–25.0%) (p < 0.01), compared with October 2019. Conversely, in participants who did not perceive declining physical fitness during the COVID-19 state of emergency, PA time during the COVID-19 state of emergency (April 2020) (–20.6%) and afterward (October 2020) (–11.8%) showed a smaller decline; however, those decreased PA times were still at a significant level (p < 0.01).

Figure 1.

** statistical significance (p < 0.01) compared with October 2019

Be consistent in terminology. Use “PA time” or total “PA time”

Response

In the revised manuscript, the phrase “PA time” is used, except in the Methods section.

Location in the text

Entire manuscript

In the discussion you may write “a decline of x%” instead of “– x% median change”

Response

We revised these according to your suggestion.

Location in the text

Discussion

The current study confirmed that weekly PA time decreased from October 2019 to April 2020 by approximately 30%. This decrease in PA time improved in October 2020; however, a decline of approximately 15% was still observed. Specifically, the decrease in PA time in participants who perceived declining physical fitness during the COVID-19 state of emergency was remarkable, with declines of approximately 50% and 25% in April and October 2020, respectively.

Reviewer 3 Report

Considering the fact that, because of the pandemic, the population has decreased their fitness activity, studies like this in which it is observed- by an online survey to a really big sample- how many have changed the median total of physical activity time of the individuals are important. Thus, first of all, it is appreciated that the authors submitted this study and I would like to congratulate them for the good work you have done in the manuscript. However, several aspects need to improve to consider for publication. Below there is a list of suggestions that I believe would improve the paper's quality.

  • Introduction last paragraph: Perhaps it is necessary to report how many lockdowns, or the duration of the lockdown(s), held in Japan during that period? Because I believe that it has been different in other countries and maybe is a piece of information that future metanalysis should use. For example, here in Spain we only had one lockdown of nearly 4 months.
  • Methods 2.3: I believe that it would be more suitable to classify the groups as "Declining physical fitness perception” instead of “having a declining physical fitness” as stated at the end of the paragraph. Please check this in the whole manuscript.
  • Also, when appeared this question? At the beginning or at the end of the survey? Please include this information.
  • Statistical analysis: I notice that the authors used non-parametric tests, which is correct. But did the authors used these tests because the data were not normal (and if so, please report what test did you use) or is it because is better to use these tests with big samples? Because you can use parametric tests in big samples, even if the data are not normal, considering the central limit theorem. In any case, this is not a problem because the results would be probably the same.
  • Results: Is interesting the result depicted in figure 1: even when the individuals reported that they did not perceive a decline in physical fitness, a reduction of the median total of PA time was observed in both April and October 2020. If we consider that the “declining physical fitness" question was answered at the end of the survey, this would imply that they probably knew that the PA time was lower than October 2019, and thus is plausible to consider that there is like some sort of self-perception threshold of physical fitness status despite knowing that they were, indeed, performing less PA. This hypothesis will not add anything to the aims of this research, but maybe could be interesting for future research lines. If the authors consider this point interesting, I believe it should be included in the discussion in a paragraph of future researches or something else.
  • Discussion last paragraph: Another limitation is that the two groups were determined according to their declining physical fitness perception at the end of the period, so first, they were not divided by objective measures, and second, some results in April 2020 could be different: some individuals of the “non-declining” group, if they were asked in April, perhaps they would answer as a “declining” group.

Author Response

Manuscript ID: ijerph-1187899
Title: Associations of physical activity changes and perceived declining physical fitness during the COVID-19 state of emergency among Japanese middle-aged adults

Reviewer 3

Considering the fact that, because of the pandemic, the population has decreased their fitness activity, studies like this in which it is observed- by an online survey to a really big sample- how many have changed the median total of physical activity time of the individuals are important. Thus, first of all, it is appreciated that the authors submitted this study and I would like to congratulate them for the good work you have done in the manuscript. However, several aspects need to improve to consider for publication. Below there is a list of suggestions that I believe would improve the paper's quality.

Response

We appreciate your helpful and insightful comments. Please see our responses to your specific comments below.

Introduction last paragraph: Perhaps it is necessary to report how many lockdowns, or the duration of the lockdown(s), held in Japan during that period? Because I believe that it has been different in other countries and maybe is a piece of information that future metanalysis should use. For example, here in Spain we only had one lockdown of nearly 4 months.

Response

I appreciate your helpful comments. The state of emergency in Japan was implemented for about a month and a half, from April 7 to May 25, 2020. During the declaration of a state of emergency, the prefectural governors of the targeted areas can request that residents refrain from going out, except when it is necessary for the maintenance of daily life. Governors can also make requests or give instructions regarding school closure and the restricted use of public facilities. The introductory text has been revised accordingly.

Location in the text

Introduction

In Japan, a state of emergency was declared in seven prefectures, including Tokyo, on April 7, 2020 and extended to the entire country on April 16 [7]. On May 25, the state of emergency was lifted nationwide for the first time in about a month and a half. Although the declaration of a state of emergency is not legally binding, the prefectural governors of the target areas can request that residents refrain from going out as part of a collective effort to reduce infection, except when doing so is necessary for the maintenance of daily life.

Methods 2.3: I believe that it would be more suitable to classify the groups as "Declining physical fitness perception” instead of “having a declining physical fitness” as stated at the end of the paragraph. Please check this in the whole manuscript. Also, when appeared this question? At the beginning or at the end of the survey? Please include this information.

Response

I appreciate your helpful suggestion. We classified the groups as “perceived decline in physical fitness,” instead of “having a declining physical fitness” throughout the manuscript. Further, the revised manuscript includes detailed information regarding the placement of the question. Self-reported data about physical fitness perception during the COVID-19 epidemic were collected before asking about PA time.

Location in the text

Methods

Participants who answered “yes” to this question were classified as perceiving a decline in their physical fitness.

Methods

Incidentally, physical fitness perception during COVID-19 epidemic was asked before responding to the IPAQ in this survey.

Statistical analysis: I notice that the authors used non-parametric tests, which is correct. But did the authors used these tests because the data were not normal (and if so, please report what test did you use) or is it because is better to use these tests with big samples? Because you can use parametric tests in big samples, even if the data are not normal, considering the central limit theorem. In any case, this is not a problem because the results would be probably the same.

Response

In this study, we followed the “Guidelines for Data Processing and Analysis of IPAQ (2005)” to analyze physical activity. According to the guidelines, given the non-normal distribution of energy expenditure in many populations, it is suggested that the continuous indicator be presented as median minutes/week or median MET-minutes/week, rather than as a mean. Therefore, we conducted the analysis using non-parametric tests to assess the medians.

Location in the text

Methods

The IPAQ guidelines state that physical activity is non-normally distributed in many populations and suggest reporting medians [20]. Therefore, we conducted the analysis using non-parametric tests, as appropriate.

Results: Is interesting the result depicted in figure 1: even when the individuals reported that they did not perceive a decline in physical fitness, a reduction of the median total of PA time was observed in both April and October 2020. If we consider that the “declining physical fitness" question was answered at the end of the survey, this would imply that they probably knew that the PA time was lower than October 2019, and thus is plausible to consider that there is like some sort of self-perception threshold of physical fitness status despite knowing that they were, indeed, performing less PA. This hypothesis will not add anything to the aims of this research, but maybe could be interesting for future research lines. If the authors consider this point interesting, I believe it should be included in the discussion in a paragraph of future researches or something else.

Response

I appreciate your helpful comments. The points you suggested would be very interesting. We considered discussing them and have added related sentences in the paragraph about future research.

Location in the text

Discussion

Therefore, some individuals may believe that there is a self-perception threshold regarding one’s physical fitness status, despite any given individual’s knowledge that they were, indeed, performing less PA. In future research, these points could contribute to understanding the associations between PA levels and self-perceived fitness.

Discussion last paragraph: Another limitation is that the two groups were determined according to their declining physical fitness perception at the end of the period, so first, they were not divided by objective measures, and second, some results in April 2020 could be different: some individuals of the “non-declining” group, if they were asked in April, perhaps they would answer as a “declining” group.

Response

The revised manuscript includes details pertaining to the question and the method by which persons who perceived a decline in their physical fitness were identified. This survey used a cross-sectional design and asked respondents to recall their PA time; objective measures were not included. We included those points in the Limitations section.

Location in the text

Discussion (limitation)

In addition, the two groups were determined according to their perception of a decline in physical fitness at the end of the survey period, October 2020; because they were not divided according to objective measures, several April 2020 results might be different. For instance, some members of the “non-declining” group may have identified as members of the “declining” group, if they had been asked in April 2020.

Round 2

Reviewer 1 Report

I thank the authors for the good work of revision.

I would pointed out two point.

  1. "PA" (l.95) was already described (l.41).
  2. The description of asterisks should not be in Figure. Please move to the caption/legend statement.

best regards.

Author Response

Manuscript ID: ijerph-1187899
Title: Associations of physical activity changes and perceived declining physical fitness during the COVID-19 state of emergency among Japanese middle-aged adults

Reviewer 1

I thank the authors for the good work of revision.

I would pointed out two point.

Response

We appreciate your helpful and insightful comments. Please see our responses to your specific comments below.

"PA" (l.95) was already described (l.41).

Response

This sentence was revised. Thank you.

Location in the text

Methods

2.2. Assessment of PA

The description of asterisks should not be in Figure. Please move to the caption/legend statement.

Response

We revised Figure 1 according to your suggestion. Thank you.

Location in the text

Figure 1. Changes in PA time in April and October 2020 compared with October 2019 in participants with and without a perceived decline in physical fitness. **: statistical significance (p < 0.01) compared with October 2019.

Reviewer 2 Report

Dear Editor and Authors

The Authors have addressed most of my concerns and improved the manuscript, however I wold like to make a few more suggestions.

There is stil rom for improvement in the language

An example from the abstract

“Overall, the PA time per week decreased between October 2019 and April 2020 with a decline of 32.4% median change.”

Can be simplified to

PA time per week decreased by 32.4% between October 2019 and April 2020.

An example from the results

“Overall, there was a significant decrease in PA time in April 2020 (median [IQR], 240 [80 150 to 540]) when compared to October 2019 (355 [150 to 660]) (P < 0.001). There was also a 151 decrease in the PA time in October 2020 (300 [120 to 600]) (P < 0.001).”

Can be simplified to

PA time decreased (P < 0.001) in April 2020 (240 [80 150 to 540]) and October 2020 (300 [120 to 600]) compared to October 2019 (355 [150 to 660]).

Please insert a reference or two comparing PA measured by survey and objective measurements when you discus the validity of the PA measurements.

I do not think that you should title your paper as a study of effect when you have used a longitudinal design.

Author Response

Manuscript ID: ijerph-1187899
Title: Associations of physical activity changes and perceived declining physical fitness during the COVID-19 state of emergency among Japanese middle-aged adults

Reviewer 2

The Authors have addressed most of my concerns and improved the manuscript, however I wold like to make a few more suggestions. There is stil rom for improvement in the language

An example from the abstract

“Overall, the PA time per week decreased between October 2019 and April 2020 with a decline of 32.4% median change.”

Can be simplified to

PA time per week decreased by 32.4% between October 2019 and April 2020.

An example from the results

“Overall, there was a significant decrease in PA time in April 2020 (median [IQR], 240 [80 150 to 540]) when compared to October 2019 (355 [150 to 660]) (P < 0.001). There was also a 151 decrease in the PA time in October 2020 (300 [120 to 600]) (P < 0.001).”

Can be simplified to

PA time decreased (P < 0.001) in April 2020 (240 [80 150 to 540]) and October 2020 (300 [120 to 600]) compared to October 2019 (355 [150 to 660]).

Response

We revised the sentences according to your suggestion. Thank you very much.

Location in the text

Abstract

Overall, the PA time per week decreased by 32.4% between October 2019 and April 2020.

Results

Overall, the PA time decreased (p < 0.001) in April 2020 (median [IQR], 240 [80 to 540]) and October 2020 (300 [120 to 600]) compared to October 2019 (355 [150 to 660]).

Please insert a reference or two comparing PA measured by survey and objective measurements when you discus the validity of the PA measurements.

Response

We added a reference and sentence in the Discussion section.

Location in the text

Discussion

Subjective PA levels may be over- or underestimated [38].

I do not think that you should title your paper as a study of effect when you have used a longitudinal design.

Response

I appreciate your comments. We reconsidered the title of this study.

Location in the text

Title

Physical activity and perceived physical fitness during the COVID-19 epidemic: a population of 40–69-year-olds in Japan

This manuscript is a resubmission of an earlier submission. The following is a list of the peer review reports and author responses from that submission.